# The Prevalence of Genital Mycoplasmas and Coinfection with *Trichomonas vaginalis* in Female Patients in Vienna, Austria

**DOI:** 10.3390/microorganisms11040933

**Published:** 2023-04-03

**Authors:** Ina Hoxha, Iwona Lesiak-Markowicz, Julia Walochnik, Angelika Stary, Ursula Fürnkranz

**Affiliations:** 1Institute for Specific Prophylaxis and Tropical Medicine (ISPTM), Centre for Pathophysiology, Infectiology and Immunology Medical University of Vienna, 1090 Vienna, Austria; 2Pilzambulatorium Schlösselgasse, Outpatients Centre for the Diagnosis of Venero-Dermatological Diseases, 1210 Vienna, Austria

**Keywords:** *Trichomonas vaginalis*, *Mycoplasma hominis*, *Candidatus Mycoplasma girerdii*

## Abstract

*Trichomonas vaginalis* causes trichomoniasis, the most recurrent sexually transmitted infection (STI) worldwide. Genital mycoplasmas, not considered STI agents, are frequently isolated from the female genital tract. A symbiosis between *Mycoplasma* species and *T. vaginalis* has been described. The aim of this study was to conduct molecular-based analyses of vaginal specimens, thus assessing the prevalence of non-STI *Mycoplasma* infections. In total, 582 samples from female patients and an additional 20 *T. vaginalis* isolates were analyzed by PCR using *Mycoplasma* specific 16S rRNA primers, and the obtained PCR products were sequenced. *Mycoplasma* species were detected in 28.2% of the collected vaginal samples. *Mycoplasma hominis* was found in 21.5% of the specimens, *Ureaplasma* species were found in 7.5% of the samples. The molecular data of the newly described species, *Candidatus*
*Mycoplasma girerdii*, were obtained for the first time in Austria, in a sample also positive for *T. vaginalis*. Analyses of the cultivated *T. vaginalis* strains confirmed the presence of *M. hominis* in two out of 20 samples. A comparably high prevalence of genital mycoplasmas was revealed through advanced diagnostic assays, with *M. hominis* and *U. parvum* being the most prevalent species. The previously described symbiotic relationship between *M. hominis* and *T. vaginalis* was confirmed.

## 1. Introduction

*Mycoplasma hominis*, *Ureaplasma urealyticum,* and *Ureaplasma parvum* are often isolated from the human genital tract, and their presence correlates with age, sexual activity, pregnancy, and hormones; however, all three species are not considered classical sexually transmitted infectious agents [1,2]. *M. hominis* is a frequent isolate from the lower genital tract of women and is thought to play a minor role in urogenital tract disease; however, it can be associated with pelvic inflammatory disease (PID), bacteremia, and bacterial vaginosis (BV) [1].

BV, previously known as ‘*Gardnerella* vaginitis’, is a condition characterized by depletion of the normal *Lactobacillus* population. As a result, it is accompanied by an overgrowth of vaginal anaerobes and the loss of vaginal acidity [3]. Overall, half of the registered cases of BV are either asymptomatic, or women exhibit only mild symptoms. The concentration of certain bacteria including *Prevotella* spp. (*Bacteroides*), *Gardnerella vaginalis*, *M. hominis,* and *Mobiluncus* spp. is significant [3,4]. *M. hominis* and antibodies against the bacterium are detectable in the vagina of approximately 60% of women diagnosed with BV. The mechanism as to how *M. hominis* contributes to BV pathology is yet to be understood [3].

*Ureaplasma* spp. are the most frequent genital tract colonizers in females. Unlike *U. parvum*, *U. urealyticum* is identified predominantly in women experiencing reproductive morbidities. Currently, there is conflicting evidence regarding the pathogenic profile and medical relevance of these two species [5]. 

Recent studies on the vaginal microbiome identified a new *Mycoplasma* species, formerly known as Mnola [6] and renamed *Candidatus*
*Mycoplasma girerdii*, which shows a strong and unique association with *Trichomonas vaginalis* [7] and has been proven recently to be an intracellular symbiont of *T. vaginalis* [8]. The bacterium is not cultivable in *Mycoplasma*-specific medium and has a small genome, lacking the genes essential for mycoplasma energy metabolism [7]. 

*T. vaginalis* is a human-specific sexually transmitted protozoan known as the causative agent of trichomoniasis. It causes over 220 million cases each year and is considered the most prevalent nonviral sexually transmitted infection worldwide [9]. The symbiotic relationship between *M. hominis* and *T. vaginalis* has been documented [10,11]. The bacterium can adhere to or enter into and proliferate inside trichomonad cells, aided by their shared property of metabolizing arginine [8,10,11]. This study investigated the prevalence of non-STI genital mycoplasmas and *T. vaginalis* infections in women attending the Outpatients Centre for the Diagnosis of Venero-Dermatological Diseases (OCD) in Vienna.

## 2. Materials and Methods

### 2.1. Study Design and Sample Collection 

Vaginal swab specimens from 582 female patients attending the OCD were collected between April and September 2021 and analyzed. Vaginal swabs were obtained during routine pelvic examinations, and no repeated samples were taken. The vaginal swab material was transferred into 2 mL tubes containing phosphate-buffered saline (PBS) and was brought to the Institute for Specific Prophylaxis and Tropical Medicine (ISPTM) for DNA isolation. The QIAamp^®^ DNA Mini Kit 250 (QIAGEN, Hilden, Germany) was used for the extraction of the DNA from the clinical samples, and the DNA was stored at −20° until further use. The *Mycoplasma* 16S rRNA gene fragment (717 bp) was amplified with conventional PCR using the GPO-1 (ACTCCTACGGGAGGCAGCAGTA) and MGSO (TGCACCATCTGTCACTCTGTTAACCTC) primers [12]. The PCR products were analyzed on 2% agarose gel using GelRedTM Nucleic Acid Gel Stain (Biotium, Hayward, CA, USA). The gene fragments were purified after visualization, using the IllustraTM GFXTM PCR DNA and Gel Purification Kit (GE Healthcare, Hatfield, UK). The purified DNA products were sequenced bidirectionally with the same set of primers, performed with the Applied Biosystems SeqStudio Genetic Analyzer (Thermo Fischer Scientific, Waltham, MA, USA). The consensus sequences were generated with the Prabi (https://doua.prabi.fr/software/cap3; accessed on 3 February 2022) Software, and were compared to the available sequences of genital mycoplasmas in GenBank using the basic local alignment search tool BLAST search (https://blast.ncbi.nlm.nih.gov/Blast.cgi; accessed on 18 January 2022).

### 2.2. Trichomonas vaginalis Samples from Pure Cultures

Twenty cultured *T. vaginalis* samples previously collected from male and female patients attending the OCD were investigated for the presence of symbiotic *Mycoplasma* species. The vaginal and urethral swabs were smeared onto *T. vaginalis* specific agar plates [13]. After microscopic observation for the presence of *T. vaginalis* at the OCD, positive samples were transferred to the ISPTM. Subsequently, the samples were suspended into liquid TYM Medium (Trypticase-Peptone Medium) and cultured microaerobically at 37 °C. The QIAamp^®^ DNA Mini Kit 250 (QIAGEN, Hilden, Germany) was used to isolate DNA from the pure *T. vaginalis* cultures. To detect the intracellular/membrane-associated genital mycoplasmas, the MGSO/GPO-1 primers were used, and the PCR products were visualized and subsequently sequenced as described.

## 3. Results

### 3.1. Data Collection and the Characteristics of the Vaginal Swab Specimens

The women screened were aged 18–93 years (mean age 35.5), and the women positive for genital mycoplasmas were aged 22–63 years (mean age 32.9). For every sample, two swab specimens were collected from each patient, one of which was smeared on an agar plate for the detection of *T. vaginalis,* and the second was subjected to molecular analysis. 

The vast majority of the patients attending the Clinic reported genital tract infection symptoms, discharge, or pain. Routine diagnostics revealed *Candida* spp., *Ureaplasma* spp., *Chlamydia trachomatis*, *Neisseria gonorrhea, Gardnerella vaginalis*, and *Prevotella* spp. The Amine Test was also performed for each specimen and if positive, together with an elevated vaginal pH as well as the presence of clue cells in the microscopic examination, this would be suggestive of BV, based on the Amsel criteria [14]. In total, 26 samples out of 582 patients fit the criteria of a BV diagnosis, yielding a positive rate of 4.4% (Table 1). 

### 3.2. The Prevalence of Mycoplasma spp. in the Swab Specimens

Overall, non-STI genital mycoplasmas were detected in 28.2% (164/582) of the samples. The most prevalent species was *M. hominis* 125/582 (21.5%) followed by Ureaplasmas: *U. parvum* 22/582 (3.8%) and *U. urealyticum* 16/582 (2.7%). No *M. penetrans* or *M. fermentans* were detected. *T. vaginalis* was detected in five specimens. In 4/5 (80%) of the specimens positive for *T. vaginalis*, *M. hominis* was also detected in the same patient. Lastly, *Ca*. *M. girerdii* was detected in 1/5 (20%) of the vaginal swabs positive for *T. vaginalis* (Table 2). 

Representative examples of the obtained *Mycoplasma* spp./*Ureaplasma* spp. sequences were submitted to GenBank and are available under the following accession numbers: *M. hominis* (OP684355-OP684359), *U. parvum* (OP684677-OP684679), *U. urealyticum* (OP684935), and *Ca*. *Mycoplasma girerdii* (OP402867). 

### 3.3. The Detection of Intracellular/Membrane-Associated Mycoplasma spp. in T. vaginalis Pure Culture

In total, 20 *T. vaginalis* isolates (eight from men and twelve from women), that had been passaged in TYM for three times, were analyzed by PCR for the detection of Mollicutes. Two samples 2/20 (10%), closely associated with *M. hominis*, were confirmed after successful amplification with the species-specific primers. *Ca*. *M. girerdii* and *Ureaplasma* spp. were not detected.

## 4. Discussion

Among the 582 swab specimens investigated, roughly 28% were positive for non-STI mollicutes by PCR. *M. hominis* infections (21.5%) were significantly more prevalent than infections with *Ureaplasma* species taken together: 7.5%. The women positive for *M. hominis* infection (mean age 32.9) were mostly of reproductive age. An earlier study [15] confirmed a correlation between sexual activity and colonization with genital mycoplasmas. Although the mean age is not an indicator of the hormonal state, an association between the latter and the isolation of genital mycoplasmas in the female genitourinary tract has also been described [16]. In that study, the occurrence of *U. urealyticum* was low among women who were sexually inactive. The highest incidence of genital mycoplasmas was observed in pregnant women, followed by sexually active nonpregnant women [16]. Several studies on female patients in Asia have reported a higher prevalence of ureaplasmas than of *M. hominis*, with 30.8% and 1.2%, respectively [17]. Although the findings in the current study do not fully align with the data from other studies related to the prevalence, a similar distribution of infections has been reported [18], indicating a lower prevalence of *U. urealyticum* compared to *U. parvum*. These previous reports further corroborate the known discrepancy in the prevalence of non-STI mycoplasmas in women and the difference in the methodological approaches implemented [13,19]. Furthermore, there is a lack of case-control studies, as a means to correlate genital *Mycoplasma* spp. infection with disease development [19]. It is noteworthy that in this retrospective study the expected overall prevalence of genital mycoplasmas would be seemingly higher for *M. hominis* and *Ureaplasma* spp. due to the higher susceptibility to infection in women [5]. 

The infection rate of female patients diagnosed with Bacterial Vaginosis (BV) was 4.4%. In all patients, elevated levels of *Prevotella* spp. and *G. vaginalis* were detected, whereas *M. hominis* was detected in 23% of the total confirmed cases of BV. A study in the USA conducted over the span of four years revealed the prevalence of BV in women under the age of 50 years old to be 29.2% [20]. Although *M. hominis* alone is not able to instigate BV, its presence correlates with a depletion of *Lactobacillus* species [21,22]. In a study from Northern Ireland, high *M. hominis* and *G. vaginalis* coinfection rates were detected, whereas the same was not observed for *M. genitalium* or *Ureaplasma* spp [23]. The current study provides data on the correlation of BV prevalence as well as the bacteria indicative of the condition and emphasizes the need for additional diagnostics to explain the underlying risk factors. 

In a 2018 statement from the European (STI) Guidelines Editorial Board, a routine-based testing of asymptomatic and symptomatic patients for *U. parvum*, *U. urealyticum,* and *M. hominis* was not recommended [19]. *M. hominis* and other genital Mollicutes are usually seen as commensals of the genitourinary tract and are detected in symptomatic as well as in asymptomatic patients. Their occurrence does not always result in discomfort or typical STI symptoms and thus they are not considered (solely) to be responsible for disease [1,5]. On the other hand, in a retrospective study conducted at the OCD [24], *M. hominis* was the only causative agent detected in male and female patients, some of whom reported symptoms of urogenital infections. This led to the assumption that the bacterium was exclusively a responsible causative agent. The introduction of multiplex PCR assays has facilitated the detection of STI agents also allowing for *Ureaplasma* spp. and *M. hominis* identification. In contrast to the current study, in routine diagnostics, genital mycoplasmal infections are mainly diagnosed by culture or culture rapid kits [13,19]. The latter have unsatisfactory specificity and sensitivity, which can lead to a clinically inaccurate interpretation of the etiological status of *U. urealyticum* and *U. parvum.* Here, we suggest that if an infection with genital mycoplasmas is suspected, laboratory diagnostics should be performed in the best manner possible. Moreover, the early data and ongoing research on genital mycoplasmas as important contributors to reproductive difficulties should not be overlooked. 

The DNA of *Ca.*
*Mycoplasma girerdii* was detected and amplified from a patient also positive for *T. vaginalis*. To our knowledge, our sequence represents the first molecular data on this bacterium in Austria to date. An interdependent relationship between *Ca*. *M. girerdii* and *T. vaginalis* was confirmed by Fettweis et al., and the adherent presence of *Ca*. *M. girerdii* at the time was not proven [7]. In the current study, coinfections with *Mycoplasma* species for all three samples positive for *T. vaginalis* were confirmed; however, the differentiation between the intracellular and adherent nature of the presence of M. hominis and *Ca*. *M. girerdii* was not investigated. In accordance with this assertion [6,25], that study also reported that the unculturable bacteria could be detected in association with *T. vaginalis* but not in pure protozoan cultures. Very recently, Margarita et al. confirmed the endosymbiotic nature of *Ca*. *M. girerdii* and its ability to multiply within trichomonad cells [8]. 

With respect to the assessment of the symbiotic relationship between *Mycoplasma* spp. and *T. vaginalis*, the presence of *M. hominis* was confirmed in all four vaginal swabs positive for *T. vaginalis*, as well as in 10% of cultured *T. vaginalis*. In Italy, a strong association between these two microorganisms was found in 78.6% of all *T. vaginalis*-positive samples [26]. A study from the Netherlands detected *M. hominis* in 79% of the *T. vaginalis-*confirmed isolates [27], whereas in the USA a lower prevalence of *M. hominis* in *T. vaginalis* (20%) was reported [28]. The two microorganisms can separately establish long-term infections in the genitourinary tract [29]; however, it has been corroborated that the pathogenicity of both is greatly affected by their symbiotic relationship [30]. Xiao et al. suggested that *M. hominis* symbiosis with *T. vaginalis* may confer better resistance against metronidazole in vitro [31]. Fürnkranz et al. found that *T. vaginalis* strains infected with *M. hominis* in vitro exhibited a twofold increase in the minimal inhibitory concentration (MIC) to metronidazole with concomitant alterations in the expression levels of the genes correlated with metronidazole resistance [32]. Hence, it may be inferred that *M. hominis* may influence the drug susceptibility of *T. vaginalis*. However, the sensitivity to metronidazole was not assessed in the present study. Many clinical and in vitro studies have confirmed the establishment of long-term infections with these two microorganisms separately, and the pathogenicity and resistance against metronidazole of the protozoan is greatly affected by their symbiotic relationship [10,31,32]. In conclusion, our findings provide an insight into the prevalence of genital mycoplasmal infections among female patients in Vienna, Austria, as well as an assessment of the endosymbiotic relationship between *Mycoplasma* spp. and *T. vaginalis*. The reported findings of *Ca*. *M. girerdii* are still very recent, and the knowledge regarding its pathogenic potential is limited. This study underlines the importance of implementing advanced molecular diagnostic techniques, thus enabling a better screening of patients and further elucidating the role of *Mycoplasma* spp., *Ureaplasma* spp., and *T. vaginalis* in sexually transmitted diseases. 

## Figures and Tables

**Table 1 microorganisms-11-00933-t001:** The characteristics of positive cases from patients diagnosed with BV.

BV	MH *	Mean Age
26 (4.4%)	6 (23%)	30.5 (20–48)

* *M. hominis*-positive samples in BV positive cases (women).

**Table 2 microorganisms-11-00933-t002:** The characteristics of *Mycoplasma*/*Ureaplasma* spp. and *T. vaginalis* positive cases in women.

	Total Samples (*n* = 582)	
*Mycoplasma* Positive Rate		*T. vaginalis* Positive (*n* = 5)
*M. hominis*	125 (21.5%) *	4 (80%) **
*Ca*. *M. girerdii*	1 (0.17%) *	1 (20%) **
*U. urealyticum*	16 (2.7%) *	–
*U. parvum*	22 (3.8%) *	–

* Percentage of the total samples. ** Percentage of the *T. vaginalis*-positive samples.

## Data Availability

Not applicable.

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
