# Peer review of "The Prevalence of Genital Mycoplasmas and Coinfection with Trichomonas vaginalis in Female Patients in Vienna, Austria"

_microorganisms, 2023, doi:10.3390/microorganisms11040933_

Round 1

Reviewer 1 Report

Congratulations to a well conducted observational study. As far as I understand the specimen were analyzed retrospectively and the anamnesis of each individual patient is missing (as it is the diagnosis of referral) so of course, there is potential some kind of selection bias inherent. However, the results are interesting, particularly for clinically working gynecologists, and considering the spare literature, I would definitively recommend to publish the work in its current form.

Author Response

Dear Reviewer!

Thank you very much for your comments, we apprechiate it very much.

best regards,

Ursula Fürnkranz

Reviewer 2 Report

Comments

Prevalence of genital mycoplasmas and co-infection with Trichomonas vaginalis in female patients in Vienna, Austria

This manuscript is a statistical survey of women infected with T. vaginalis who concurrently have Mycoplasmas on gynecological examination tests in Vienna, Austria.

Despite being well described, the authors did not conduct an adequate bibliographic survey. As a result, many references are missing, and the discussion is thus poor. Furthermore, they forgot to mention the importance of mycoplasmas inside T. vaginalis, with a consequent increase in the pathogenicity of the infection, as previously demonstrated in good articles in the literature.

They also do not present a single image or table of their results, which is important in a scientific publication.

They should also present data from patients who concomitantly present mycoplasma and T. vaginalis exposing their symptoms. Thus, without this information, the manuscript will only be descriptive.

Minor comments

Lines 45-46- “The symbiotic relationship....” -Please, add references.

Line 72: “20 cultured..” Please, change to Twenty

Lines 103-104: “ Two samples 2/20 (10%) harbored M. hominis (intracellularly or attached to the cell membrane”- This referee wants to see images of this observation.

Round 2

Reviewer 2 Report

No comments. The authors made the corrections.